# Management of Bone Health Considerations in Patients with Cancer

**DOI:** 10.3390/cancers17172878

**Published:** 2025-09-01

**Authors:** Michelle Brennan, Tania Kalsi

**Affiliations:** 1Department of Geriatric Medicine, University Hospital Limerick, V94 F858 Limerick, Ireland; 2Department of Ageing and Health, Geriatric Guy’s and St Thomas’ NHS Foundation Trust, London SE1 9RT, UK

**Keywords:** ageing, bone health, cancer treatment-related bone loss, fracture risk, geriatric oncology, osteoporosis, survivorship

## Abstract

Older adults, accounting for the majority of new cancer diagnoses, are living longer with cancer due to more effective treatments and earlier detection. Osteoporosis is a disease which affects the bones and makes them more fragile, leading to fractures. Osteoporosis and broken bones are a risk in older people with cancer due to increasing age itself as well as an impact of the drug treatments and the radiation given to treat cancer. Older people with cancer should have a personalised bone health assessment as part of their routine cancer care and be educated on lifestyle changes that can prevent fractures. Those at risk should be offered a DEXA scan, which looks at the strength (density) of their bones and offered medications to reduce their chances of fracture if they have low bone density or previous fractures. While there are guidelines for certain patient groups, like those on hormone treatments and steroids, there is a need for updated guidelines to include newer cancer treatments. Strategies to reduce the impact of cancer treatment on bones and the monitoring of bone health should be included in long-term cancer survivorship programmes.

## 1. Introduction

Given the global ageing population, cancer among older people is a significant health burden that is expected to grow over the next few decades. People aged 65 years or older account for more than 60% of cancer diagnoses [1].

Older people have higher mortality rates due to cancer, with over half of all cancer deaths occurring in those aged 75 years or older [2]. However, the age-standardised mortality rates are falling each year due to a number of improvements, including earlier diagnoses through screening programmes for some cancer types, and the advent of new advances in cancer treatments such as immune checkpoint inhibitors, which are highly effective in many advanced and metastatic cancer settings [3].

The number of people living with a cancer diagnosis in the UK is growing rapidly and is estimated to exceed 4 million by 2040 [1]. Cancer survivorship will form an integral part of future health care needs for the older population. Health services must be equipped to support recovery following a cancer diagnosis and treatment but also focus on sustaining health and wellbeing, whether during cancer survivorship or as part of managing cancer as a chronic disease. The NHS Improvement Plan for Cancer Survivorship recommends care models that consider the specific disease, treatment received and its effects on the person, and each individual’s personal set of circumstances [4]. Holistic approaches to cancer care and cancer treatment optimisation are recommended by international bodies such as the International Society of Geriatric Oncology (SIOG) and the American Society of Clinical Oncology (ASCO) using a comprehensive geriatric assessment (CGA) approach [5,6]. CGA is a vehicle for the delivery of holistic patient-centred care and incorporates a wider needs assessment, including screening and the management of geriatric syndromes. One common geriatric syndrome that is increasingly in need of a proactive approach in cancer care is osteoporosis.

Older people are already at an elevated risk of osteoporosis and fracture. In the UK, there are 3.7 million people diagnosed with osteoporosis, and the vast majority are female. Over half a million fragility fractures occur each year, with an approximate cost of 2.4% of all healthcare spending. It is associated with higher morbidity (e.g., pain and disability) as well as some fractures being associated with premature mortality [7]. The annual number of reported deaths due to osteoporosis in the UK is 114,000, which is higher than the average annual mortality rate of lung, bowel, prostate, breast, pancreas, oesophageal, liver, bladder and ovarian cancer combined [8].

Cancer is a major risk factor for accelerated bone loss and fracture risk. Whilst adverse bone health in cancer survivors with a history of multiple myeloma, prostate cancer and breast cancer has been established and published guidelines exist for the treatment and management of bone health in these populations, there is little known about bone health in other cancer types. A recent population-based matched cohort study using UK electronic healthcare records found that, compared with cancer-free individuals, the risk of any bone fracture was increased in fifteen out of twenty common cancer types and major osteoporotic fracture risk was increased in seventeen out of twenty cancers [9]. The risk of fracture persisted over five years in several cancer types, although the strength of association appeared to reduce over time since the cancer diagnosis. The risk to bone health in patients with cancer is multifactorial, and variations in patterns of observed fracture risk indicate the presence of distinct mechanisms as well as some shared risk factors.

There is evidence of a direct effect of cancer cells on bones due to local cellular activation, which is particularly evident in haematological malignancies and direct invasion of bone in those with bone metastasis. Increased risk in breast and prostate cancer survivors may be related to a higher tendency to develop metastatic disease in the spine and pelvic areas, combined with the impact of hormonal therapies in suppressing sex hormones and the subsequent impact on bone mineral density loss [10]. Additionally, cancers that affect balance or motor control, such as those affecting the central nervous system, spine or lower limbs, could result in gait impairment and thus increase the frequency of falls.

Oncology services need to be cognisant of the impact of cancer on bone health and implement strategies to minimise bone loss and reduce fracture risk as part of long-term cancer care and survivorship programmes in all cancer types. This review paper discusses the effects of cancer and cancer treatments on bone loss, as well as strategies to optimise bone health in patients living with or surviving cancer.

### 1.1. General Factors and Risk of Secondary Osteoporosis in Cancer Populations

As a result of the normal aging process, the bone deteriorates in composition, structure and function, which predisposes to osteoporosis and fracture. The predominant mechanism for age-related bone loss is gonadal sex steroid deficiency, leading to a decline in oestrogen and testosterone levels. Women lose about 50% of their trabecular bone and 30% of their cortical bone during their lifetime, about half of which is lost during the first decade following the menopause [11].

Additional intrinsic and extrinsic factors accelerate this decline in bone mass and predispose a person to fracture. Intrinsic factors include genetics, peak bone mass attainment, alterations in cellular components, hormonal, biochemical and vasculature status. Extrinsic factors include nutrition, physical activity, co-morbid medical conditions and medication usage [12]. People living with cancer and those going through cancer treatments are at risk of weight loss and malnutrition due to the catabolic effects of cancer itself, malabsorption in the case of some gastro-intestinal or hepatobiliary malignancies, and reduced oral intake is common due to nausea and the development of oral problems such as mucositis during some cancer treatments.

Over half of older people living in the UK have two or more chronic health conditions [13]. Those living with multi-morbidity may have additional risk factors for fracture, such as reduced physical activity levels, a higher number of falls, and an increased likelihood of polypharmacy, where additional medications are prescribed for co-morbid medical conditions.

While there is currently no accepted national screening programme, the SCOOP trial in the UK found community-based screening programmes targeting women aged 70–85 years using BMD and the FRAX risk assessment tool reduced hip fractures and were found to be cost-effective [14].

### 1.2. Effect of Cancer of the Skeletal System

Almost all cancers have negative effects on the skeletal system. A prospective study of over 1000 oncological inpatients who underwent bone mineral density (BMD) testing demonstrated higher rates of bone loss in patients with cancer compared with the general population, independent of sex or cancer type [15].

In patients with non-metastatic cancer, both the disease itself, through increased local and systemic inflammatory effects, as well as cancer treatment, can pose challenges to the skeletal integrity [16]. Chronic inflammation can promote bone loss through alterations to the bone remodelling process, increased bone resorption and impaired bone formation [17]. This occurs due to the effects of pro-inflammatory cytokines such as tumour necrosis factor, interleukin-1, interleukin-6, macrophage colony-stimulating factor, and RANK ligand (RANK-L) on osteoclastic and osteoblastic activity [18].

Osteosarcopenia, characterized by the concurrent deterioration of bone and muscle mass, represents a crucial yet often underrecognized complication in people living with cancer. It not only poses significant challenges to patients’ mobility and quality of life but also has profound implications for their treatment outcomes and survival rates. A systematic review and meta-analysis of eight good-quality studies of cancer patients with or without osteosarcopenia found that having osteopenia, sarcopenia or osteosarcopenia was associated with worse disease-free survival. The study reported pooled hazard ratios of 1.70 for osteopenia and 2.17 for osteosarcopenia, respectively [19].

The ASCO 2019 guidelines on the management of osteoporosis state that patients with non-metastatic cancer may be at higher risk for osteoporotic fractures due to their baseline risk factor profile or due to the added risks that are associated with their cancer therapy [16].

Bone is one of the most common sites for metastatic spread. Worldwide, more than 1.5 million people experience bone metastases, especially from breast, prostate, and lung cancer. Due to improvements in cancer treatments and improvements in overall survival in those with metastatic disease, assessing the risk of pathological fracture is an important consideration to guide decisions around physical activity and rehabilitation goals, use of anti-resorptive therapies, and decisions regarding local interventions [20]. Specific treatment guidelines exist for the use of bone-targeted agents in the setting of bony metastasis in certain cancers and are outside the scope of this article [21].

### 1.3. Hormonal Therapies

Cancer treatment-induced bone loss (CTIBL) is an important consideration for patients receiving hormonal deprivation therapy for breast or prostate cancer.

CTIBL is the most common long-term adverse event experienced by breast cancer patients. Endocrine therapy is the standard of care for adjuvant treatment in people with hormone-sensitive breast cancer, which represents 75–80% cases. Hormonal therapy consists of two main drug classes: estrogen receptor modulators (tamoxifen) and aromatase inhibitors (anastrozole, letrozole and exemestane) [22]. Tamoxifen is not associated with significant bone loss in postmenopausal women; interestingly, there is evidence of bone protection provided by the drug and small increases in bone density may occur. Aromatase inhibitors work by blocking estrogen production and are effective breast cancer treatments in estrogen receptor-positive breast cancer; however, by dramatically reducing estrogen levels, they increase the rate of bone loss and fracture risk [23]. Estrogen deprivation leads to accelerated bone turnover with an annual bone loss of 2.2–2.6% at the lumbar spine and 1.7–2.1% at the hips. This decrease in BMD results in up to a 40–50% increase in fracture incidence attributed to the combination of cancer treatment and background post-menopausal bone density loss [24].

Androgen deprivation therapy (ADT) is a common treatment for advanced and metastatic prostate cancer, with half of patients with prostate cancer having hormonal therapy at some point during their treatment [25]. ADT suppresses estrogen and testosterone production, leading to accelerated bone loss and higher fracture risk. In a meta-analysis of fourteen trials, men who received ADT experienced a 23% increase in overall fracture risk compared with men with prostate cancer without exposure to ADT [26]. A recent meta-analysis of randomised control trials comparing fracture and fracture risks in patients treated with androgen receptor signalling inhibitors (ARSI) with standard ADT treatment demonstrated a relative risk ratio of 2.32 for fracture and 2.22 for falls in the ARSI group [27]. Bone loss is highest during the first year of ADT with an estimated BMD loss of 5–10%. This decline continues at a more gradual rate throughout the duration of ADT treatment [28].

The *ESMO Clinical Practice Guidelines* on bone health in cancer recommend that women with breast cancer receiving aromatase inhibitors and men with prostate cancer receiving ADT should have a fracture risk assessment based on the presence of clinical risk factors and BMD [21]. Bone-targeted agents are recommended if there are more than two risk factors present for fracture: a T-score of <−2.0 on BMD testing or annual bone loss on treatment exceeds 5%.

In ABCSG-18, a multicentre, randomised, double-blind, placebo-controlled trial of adjuvant denosumab 60 mg every six months in combination with calcium and vitamin D supplementation in patients with breast cancer reported a significantly delayed time to first clinical fracture with a statistically significant hazard ratio of 0.50 [29].

A pooled analysis of head-to-head studies of monthly treatment with denosumab compared with zoledronic acid in patients with different solid tumour types, including breast and prostate cancer, found that denosumab significantly delayed the time to first skeletal-related event compared with zoledronic acid in patients with solid tumours and bone metastases [30].

Denosumab was also found to have a delayed time to multiple skeletal events and pain worsening, while both drugs had a similar effect on overall survival and disease progression.

In patients with prostate cancer receiving ADT, alendronate, risedronate, pamidronate and zoledronic acid are all shown to prevent bone loss. The ESMO bone health guidelines recommend intravenous zoledronic acid at 6–12 monthly intervals or denosumab at 6 monthly intervals as the most convenient and reliable treatment options, although denosumab is the only medication for a specific licence for bone density loss associated with ADT [21]. There are some benefits of denosumab over zoledronic acid in metastatic solid tumours because of its superior efficacy in terms of delaying the time to skeletal-related events, the convenience of a subcutaneous injection and the lack of renal function monitoring.

The NCCN taskforce guidelines published in 2013 which pre-date the recent ESMO guidance recommend calcium and vitamin D supplementation with either denosumab 60 mg subcutaneously every 6 months, zoledronic acid 5 mg intravenously every year or alendronate 70 mg orally on a weekly basis for men when their 10-year risk of hip fracture exceeds 3% as calculated by the FRAX^®^ score to prevent ADT-related bone loss [31]. In aromatase-inhibitor-related bone loss, the NCCN recommends denosumab in combination with calcium and vitamin D supplementation as the preferred bone-targeted agent where possible.

### 1.4. Chemotherapy Agents

Chemotherapy can lead to bone loss due to indirect systemic effects, the most described being loss of ovarian function in pre-menopausal women with subsequent rapid bone loss. Chemotherapy also has a direct impact on bone remodelling, leading to increased bone resorption and a decrease in BMD [32].

A prospective study in postmenopausal women with breast cancer demonstrated a significant association between cytotoxic chemotherapy and worsening of BMD with an increase in FRAX scores. At six months, there was a 2% reduction in BMD and a 1% increase in ten-year major osteoporotic fracture risk measured using FRAX^®^ seen in this study population [33]. Additional studies evaluating adjuvant chemotherapy in premenopausal breast cancer consistently reported a decrease in BMD during the first year following initiation of therapy [34].

Potential mechanisms for bone loss vary among different chemotherapy drug classes. For example, cyclophosphamide and methotrexate are thought to directly affect bone metabolism by inhibiting bone remodelling. 5-fluorouracil can induce trabecular bone loss due to enhanced bone resorption, leading to suppression of cell proliferation and promotion of apoptosis of chondrocytes and osteoblasts [33]. Taxane chemotherapies can result in myelosuppression with a subsequent increase in bone resorption, which was shown to be associated with increased expression of monocyte chemoattractant protein 1 and other inflammatory cytokine activity [35].

There is a paucity of evidence to recommend bone-targeted agents for non-metastatic cancer undergoing chemotherapy. The ESMO guidelines suggest zoledronate with GnRH analogues during neo-adjuvant chemotherapy for women with early breast cancer as a low-grade recommendation [21]. International guidelines, including ASCO and SIOG, recommend screening and performing serial monitoring of BMD in patients with breast cancer and high-risk features for osteoporosis [16]. There are a few published randomised trials on the use of bone-targeted agents in lung cancer and other solid tumours with bone metastasis. One placebo-controlled trial of zoledronic acid in cancers other than breast or prostate found that zoledronic acid reduced the number of skeletal-related events and time to first event [36].

### 1.5. Radiotherapy

Radiotherapy accounts for a major component of cancer treatments. Based on evidence-based indications, over half of cancer patients should receive at least one radiation treatment during their disease course, either alone or in combination with other treatments [37]. While technological advances have improved the accuracy and safety of radiotherapy delivery, long-term toxicities remain an important part of cancer survivorship.

Specific bone complications of radiation exposure include osteopenia, growth arrest, fracture and malignancy. While the exact pathogenesis is not understood completely, there is a consensus that radiation decreases the number of active osteoblasts by arresting these in the cell cycle, altering their ability to differentiate, and sensitizing the cells to apoptosis [38].

Bone toxicity, and more specifically radiotherapy-related insufficiency fractures (RRIFs), are well-established late effects of pelvic radiotherapy, with significant implications for mobility, morbidity and quality of life in this patient group [39]. Insufficiency fractures tend to affect bones under the most physiological stress and with the highest ratio of trabecular to cortical bone.

A recent meta-analysis of almost 4000 women who underwent pelvic radiotherapy for gynaecological malignancies reported a 14% rate of pelvic insufficiency fractures [40]. Many published studies had flaws in their study design, including retrospective methods of data collection and insufficient information around the measurement of BMD and objective fracture risk assessments. One prospective study found an increase in the proportion of patients with osteopenia or osteoporosis at three months, one year and two-year intervals following radiotherapy, when compared to baseline BMD [41]. Osteoporosis and low BMD prior to or following radiotherapy have been suggested as risk factors for RRIFs. Further studies are required to determine how fracture risk should be best assessed and managed in patients receiving radiotherapy and to explore the value of RRIFs in predicting future RRIFs and other osteoporotic fractures.

The data regarding pathologic fractures following radiotherapy is controversial. A systematic review of pathological fractures occurring after radiotherapy reported incidence rates varying from 1.2% to 25%; however, most included studies were of low-quality evidence. Fractures were found to be more frequent in the ribs in patients receiving radiotherapy for breast cancer, in the pelvis in patients treated for cancer within abdominal and pelvic organs, and in the femurs of those treated for soft tissue sarcomas [42]. In many cases, radiotherapy for skeletal metastasis is performed as a palliative measure for symptomatic relief and uses relatively low-dose radiation [43]. In these circumstances, the benefits experienced in terms of pain relief in patients with limited life expectancy are likely to outweigh the risks of radiotherapy-related fracture in the short term.

Anti-resorptive drugs such as bisphosphonates can be used to treat radiation-induced osteoporosis on a case-by-case basis, although the incidence of RRIF is better characterised than intervention efficacy in the available literature. A prospective randomized control trial, RadBone, comparing the feasibility and acceptability of a multi-modal musculoskeletal health package, including bisphosphonates for high-risk individuals and in women undergoing pelvic radiotherapy over an 18-month period, is ongoing [44].

Risedronate, zoledronic acid and parathyroid hormone have been studied in mouse models and have been found to be effective agents [45,46]. There is currently no standardised approach for the management of radiotherapy-induced osteoporosis or insufficiency fracture amongst cancer services and it merits further research and the development of clinical guidelines.

### 1.6. Glucocorticoid-Induced Osteoporosis

Glucocorticoids (GC) are administered as part of many cancer treatment regimes. In haematological malignancies, GCs (most commonly dexamethasone) are routinely included in chemotherapy protocols to induce apoptosis [47]. In non-haematological malignancies, GCs may be used as monotherapy or combined with chemotherapy or hormonal therapy in breast and prostate cancer. Additionally, GCs are used for therapeutic effects to prevent chemotherapy-induced emesis, increase appetite, improve fatigue and for specific anti-inflammatory effects, such as when used in CNS malignancies [48].

Glucocorticoid-induced osteoporosis is the most common secondary cause of osteoporosis. Rapid bone loss occurs following initiation of GCs in the first three to six months and continues to decline at a slower rate with continued use [49]. Fragility fractures, particularly vertebral fractures, occur in 30 to 60% of adults receiving long-term GC [50].

The NOGG guideline for osteoporosis in general populations, recommends that bone protective treatment should be started without waiting for BMD values in those at high risk of fracture; people with prior fragility fracture, women aged more than seventy years, postmenopausal women and men aged more than fifty years, who are either receiving high dose of GCs (>/= 7.5 mg prednisolone or equivalent daily for more than three months) or have a FRAX^®^ score exceeding the intervention threshold. People who do not meet these criteria should be assessed using FRAX^®^ with BMD assessment, where available, to guide treatment decisions. The fracture probability should be adjusted according to whether they are receiving a low, medium or high daily dose of glucocorticoids [51].

Patients who are taking GCs for cancer treatment are often prescribed intermittent courses of high-potency steroids such as dexamethasone, which can result in very high cumulative steroid dosing, particularly when repeated over multiple treatment cycles. Table 1 and Table 2 provide examples of steroid equivalency calculations of commonly used regimens.

Current oncological and osteoporotic guidelines do not provide any specific recommendations for patients receiving intermittent courses of high-dose GC. A British Society of Haematological Good Practice Paper on GC usage as a treatment for immune thrombocytopenia in non-cancer populations recommends that adults aged more than forty years who are receiving high-dose intermittent steroids are at risk of fragility fracture when the cumulative dose exceeds 1000 mg [52]. A cumulative dose exceeding 5000 mg prednisolone is associated with a relative risk increase of 14.4 for fragility fracture in one study [53]. There is limited study evidence on the effect of lower cumulative doses and cohort studies to date have not found a significant increase in fracture risk at doses under 1000 mg [54]. There is an urgent need for prospective studies to evaluate the impact on fracture risk in patients with cancer prescribed high-dose intermittent GC.

### 1.7. Immunotherapy

Immune checkpoint inhibitors (ICI) have revolutionised the way we treat many cancers over the last decade. They are now established as a powerful treatment option for advanced and metastatic cancer and have been shown to be very efficacious with a favourable toxicity profile in older adults compared to traditional cytotoxic regimes [55]. With widespread usage and improvements in overall survival for many cancer types, consideration must be given to immune-related adverse events and long-term toxicities. The impact of immunotherapy on the skeletal system is an evolving area of research with evidence emerging from animal and human studies. Two small retrospective case series of bone-related immunotherapy events reported new fracture events following initiation of ICI [56,57] (GRADE C level of evidence).

Three individual studies reported on skeletal events collected for pharmacovigilance analysis in the FDA Adverse Event Reporting System (FAERS) database, GRADE C level of evidence [57,58,59]. Pundole et al. reported that several reports of skeletal AE were collected, especially fractures, seldom reported before. Several safety signals for different types of fractures were detected in relation to all the investigated ICIs [58]. Filipini et al. reported 650 patients (0.68%) with bone and joint injuries where at least one ICI was a suspect agent from 2024 to 2020 [57]. A further analysis by Liu et al. using the same dataset over an eleven-year period, 2010 to 2021, examining 292,378 adverse events where an ICI was a suspect agent, reported 1010 reports of ICI-induced fractures [59]. Spinal, hip and femoral fractures were the most prevalent and falls were frequently reported co-morbidity in those with fractures.

A retrospective study of 1600 ICI users on the Alberta Cancer Registry reported a 2.43 calculated incidence rate ratio of sustaining a major fracture in the year following ICI initiation compared to pre-initiation, which was clinically significant in this patient population [60]. Pre-clinical and clinical studies examining potential mechanisms of immune-related bone events suggest ICIs disrupt bone remodelling and increase osteoclastic activity, which merits further investigation given the expanding use of these agents in many cancer types [61].

## 2. A Practical Summary Approach to Bone Health in Cancer Patients

### Fracture Risk Assessment in Patients with Non-Metastatic Cancer

Patients with cancer who have any additional risk factors listed in Table 3 below are considered to be at increased risk for developing osteoporosis and increased fracture risk. All patients should undergo a full clinical assessment, including history and an examination to screen for additional risk factors. Vigilance should be maintained for other causes of secondary osteoporosis where the history is suggestive or where clinical suspicion exists.

The ASCO group recommends that clinicians use a risk assessment tool such as the Sheffield FRAX^®^ tool, which should be combined with clinical judgement to interpret the results. Fracture risk assessment tools have not been validated specifically in cancer populations and thus may under-report fracture risks in this patient group, where additional risk factors exist but are not captured by the tool. To mitigate against under-treatment of an at-risk population, patients with cancer should be considered to have risk factors for secondary osteoporosis, and those receiving regular GC as part of their treatment regimen should be considered to be on regular steroids and the relevant box should be checked on the FRAX^®^ tool by the healthcare professional performing the bone health assessment.

The authors recommend following the NOGG guidelines for those receiving long-term daily steroids, either as part of their cancer treatment or for the management of co-morbid conditions.

The evidence for intermittent steroids is less clear. In the absence of robust data on the impact of intermittent GC on fracture risk in cancer populations, the authors’ believe it would be good clinical practice to consider expert opinion guidance to estimate the GC exposure as part of the proposed or previous cancer treatments and consider patients exceeding 1000 mg prednisolone to be at elevated fracture risk who may merit fracture risk assessment and consideration of bone-targeted agents.

## 3. Treatment Decisions

### 3.1. Non-Pharmacological Measures

All patients with cancer should receive education around bone health as part of their usual cancer care pathways and this should be included in survivorship programmes.

All patients living with cancer should be advised on non-pharmacological measures for optimal bone health, similar to advice for the non-cancer population. Interventions include regular exercise involving weight bearing activity and resistance training, falls prevention advice, optimising vitamin D and calcium stores through diet and/or supplementation and addressing modifiable secondary osteoporosis risk factors such as avoiding smoking and excess alcohol intake and maintaining body mass index (BMI) within the recommended range [62].

### 3.2. Pharmacological Measures

Bone-targeted agents such as bisphosphonates or denosumab are recommended for certain patients based on their risk factors, BMD assessment and presence of fragility fractures. We recommend that the ESMO clinical guidelines on bone health be followed, where it is possible to identify patients who would benefit from anti-resorptive therapy with denosumab or bisphosphonates in addition to lifestyle measures.

Health care professionals should consider patients on hormonal therapy and those receiving high-dose steroids as higher risk and thus adopt a lower threshold to treat in these settings. A shared decision-making approach should be used with the patient and their health care providers to discuss potential options and the risks and benefits of pharmacological treatment. Patients with bone metastasis and those with suspected pathological fractures should be managed using oncological best practice guidelines by their treating cancer team.

If routine access to BMD assessment with DEXA is not readily available, a FRAX^®^ assessment can be carried out, taking into account the patient’s clinical risk factors. Monitoring of BMD should occur every two years for those who remain on hormonal treatments to monitor for a decline in BMD as part of standardised monitoring for treatment toxicities.

Medication-related osteonecrosis of the jaw (MRONJ) is a potentially serious yet relatively uncommon complication of therapy with intravenous bisphosphonates and denosumab among patients with advanced malignancy, although it is more common in patients with cancer than in patients who are treated with osteoclast inhibitors for osteoporosis. A systematic review and meta-analysis of 23 randomised control trials including 42,003 patients with bone cancer metastasis [63]. This study reported that the overall ONJ incidence in cancer patients receiving denosumab or bisphosphonates was 2.08% (95% CI 1.37–2.91; *p* < 0.01). Patients receiving denosumab had a higher ONJ incidence than those receiving bisphosphonates (RR 1.64, 95% CI 1.10–2.44; *p* < 0.05).

All patients being considered for treatment with bisphosphonates or denosumab should undergo a comprehensive dental check-up with a dental surgeon, and any preventative dentistry to reduce the risk of future infection should be performed if required, prior to commencing these agents. Additional risk factors include having invasive dental procedures or concomitant oral disease increases risk, as well as trauma during intubation, fractures, dentures, preexisting inflammatory dental disease (periodontal disease, periapical pathology such as abscessed teeth), dental caries, and mandibular tori (bone exostosis). The importance of regular dental exams, ongoing dental hygiene and vigilance regarding any peri-oral symptoms in patients receiving these agents should be explained. Healthcare professionals should be cognizant of the risk of MRONJ and routinely monitor for complications while patients are receiving bone-targeted agents [64].

Several anabolic agents are now licenced for the management of severe osteoporosis in many countries and are increasing in popularity. Parathyroid hormone-related protein analogues, abaloparatide and teriparatide, are contraindicated in patients with bone metastases or a history of skeletal malignancies and those with hypercalcaemia, which can occur in the setting of malignancy.

Romosozumab, a humanized monoclonal antibody that targets sclerostin, significantly increased BMD at the lumbar spine, total hip, and femoral neck compared to placebo and active comparators in patients with primary osteoporosis [65]. However, it has not been studied to date in patients with cancer-related osteoporosis.

### 3.3. Falls Prevention

Falls are a common occurrence in patients with cancer, with fall incidence as high as 50% [66]. All older patients should undergo a falls risk assessment as part of comprehensive cancer care, with a focus on optimising mobility, balance and muscle strength to reduce the risk of falls and subsequent fractures during their cancer treatment [67].

### 3.4. Special Circumstances

A pragmatic approach should be considered for those with a very limited life expectancy of less than one year to avoid unnecessary investigation and treatment burden. A one-off dose of IV zoledronic acid can be considered for patients living with advanced frailty, who are ambulatory and deemed to be at very high falls and subsequent fracture risk, who are clinically stable at the time of their assessment and have no known contraindications to bisphosphonate therapy [68].

### 3.5. Shared Decision-Making

Shared decision-making models should be incorporated into treatment decisions and consider clinical fracture risk, expected survival, and patient preferences. Patients should have access to decision aids to provide evidence-based information about potential benefits and harms of pharmacological treatments, as well as information on individualised fracture risk. There is consistent evidence that decision support aids, when designed for patients, are beneficial and help ensure that patients have an informed choice [69]. Shared decision-making models should be adapted for patients living with severe frailty and those with limited life expectancy due to their cancer or other co-morbidities [70].

A practical clinical care algorithm is shown in Figure 1 below to summarise the above recommendations for patients with cancer.

## 4. Discussion

Management of long-term bone health is an important consideration for patients living with cancer. While expert guidelines are published for bone protection in patients with multiple myeloma and for management of cancer treatment-induced bone loss in patients receiving hormonal therapy for breast and prostate cancer, there is a lack of comprehensive guidelines addressing bone health for other cancer types [21].

While studies have reported an association between chemotherapy and radiotherapy on bone loss and fracture risk, there remains a paucity of evidence around the relative risk of fracture and the need for preventative targeted treatment interventions. The ASCO guidelines recommend that all patients with non-metastatic cancer may be at an increased risk for osteoporotic fractures due to their baseline risk factors, or additional risks acquired during their cancer treatment [16]. The ASCO advises clinicians to assess fracture risk using standardised tools and BMD assessment where indicated; however, it is important to remain cognisant that patients with cancer are likely to have additional risk factors for secondary osteoporosis that may not be captured using standardised tools. A shared decision-making discussion with the patient, considering their individualised risk profile and the overall benefits and potential harms of initiating treatment, should occur as part of routine follow-up care and survivorship models.

The ESMO guidelines for cancer-treatment-induced bone loss in patients receiving hormonal therapy outline a recommended treatment strategy using bone mineral density and additional risk factors to stratify patients [21]. Interestingly, the treatment threshold of an absolute T score of <−2.0 is lower than the standard T score of −2.5, which is the accepted definition of osteoporosis in the absence of fragility fractures. The lower threshold recommended by ESMO reflects the increased risk of fragility fracture in patients receiving hormonal therapy and the need to intervene at higher bone mineral density scores than in the general population.

Glucocorticoid-induced osteoporosis is well described, and robust guidelines exist for those on daily long-term GC regimes [51]. There is less certainty around the impact of intermittent high-dose GC, which are often used as part of cancer treatment regimes. The best available evidence comes from haematology guidelines for high doses of GC in non-metastatic conditions. In the absence of specific evidence-based guidelines for intermittent steroid regimes, all patients who are commenced on steroid treatment with an expected duration of 3 months or more should have a baseline bone health assessment performed using the FRAX^®^ tool, with steroid use indicated on the tool.

There is emerging evidence of immune-related bone events on ICIs, which, although are reported uncommonly, may increase in significance as older people are increasingly treated with these agents and maintenance immunotherapy regimes can continue for extended periods of time where they are effective in sustaining disease remission. Further research is needed to ascertain the risk of skeletal events in this population and whether immunotherapy poses an additional risk for accelerated bone mineral density loss.

All patients with cancer should be educated about the non-pharmacological management for bone health protection and undergo an individualised fall risk assessment to mitigate the risk of fragility fracture. Pharmacological treatment with bone-targeted agents is indicated in certain circumstances, with specific guidance for those on daily steroid dosing and hormonal therapy. A pragmatic approach should be considered for those with very limited life expectancy, where initiation or continuing bone-targeted treatments is burdensome for the patient and their care provider, and they are unlikely to survive long enough to benefit from the treatment effect.

## 5. Conclusions

As more patients are now surviving with cancer for much longer time periods, bone health needs to be recognised and managed as an integral part of cancer care and long-term toxicity prevention.

With increasing knowledge of the deleterious impact of cancer, cancer treatments, and their cumulative effect on baseline bone health, a targeted approach is essential to mitigate osteoporosis and fracture risk. This approach should be individualised to each person’s general risk factor profile, including their cancer-specific and cancer treatment-specific risks. This narrative review provides a broad overview of the current evidence-based guidelines and expert consensus, providing a contextual framework for future research and identifying trends in cancer-related skeletal events. This review has highlighted evidence gaps and areas of uncertainty that should be the focus of future research, including a structured systematic literature review to provide a higher level of evidence for fracture prevention within the cancer population.

## Figures and Tables

**Figure 1 cancers-17-02878-f001:**
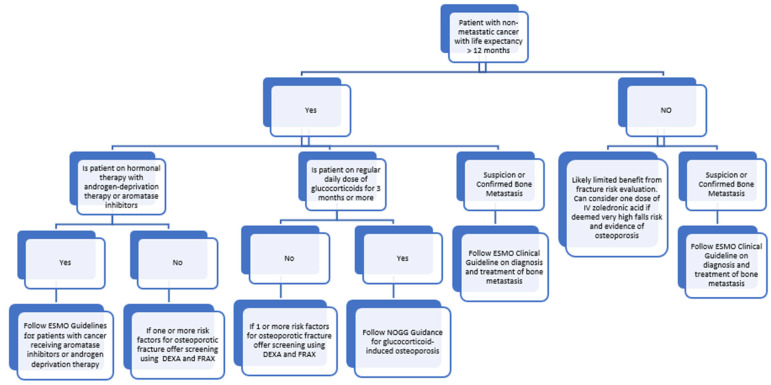
Clinical care algorithm for bone health in cancer.

**Table 1 cancers-17-02878-t001:** Cumulative GC dose for typical cancer therapy regimes.

NOGG GC Exposure	Dosing Regime	Prednisolone Equivalence
High Daily Dose	7.5 mg Prednisolone daily over 90 days	675 mg
Medium Daily Dose	5 mg Prednisolone daily over 90 days	450 mg
Low Daily Dose	2.5 mg Prednisolone daily over 90 days	225 mg

**Table 2 cancers-17-02878-t002:** Examples of commonly prescribed GC regimes in cancer treatment.

Examples of GC Regimes in Cancer Treatment		
GC Exposure	Dosing Regime	Prednisolone Equivalence
Standard dosing with chemotherapy cycles	8 mg Dexamethasone daily for 3 days	160 mg per cycle
High dosing with chemotherapy cycles	24 mg Dexamethasone day 1 of chemotherapy and 8 mg for 3 days after	320 mg per cycle
Low Daily Dose	4 mg Dexamethasone daily over 30 days	801 mg per month

**Table 3 cancers-17-02878-t003:** Risk factors for developing osteoporotic fractures.

Risk Factors for Developing Osteoporotic Fracture
Older age Current cigarette smoking Excessive alcohol consumption History of prior fragility fractures in adulthood Hypogonadism Impaired mobility History of falls or increased risks for falls Exposure to glucocorticoids for three months or more Low body weight Parental history of hip fracture Postmenopausal status

## Data Availability

No new data were created or analyzed in this study. Data sharing is not applicable to this article.

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
