# Peer review of "Management of Bone Health Considerations in Patients with Cancer"

_cancers, 2025, doi:10.3390/cancers17172878_

Round 1

Reviewer 1 Report

Comments and Suggestions for Authors

The core theme—cancer-treatment-induced bone loss and its management—is not new; authoritative guidelines already exist (ASCO 2019, ESMO 2020). The manuscript’s value lies in bringing multiple guidelines under one roof and offering a practical care algorithm that spans chemotherapy, hormonal therapy, radiotherapy, glucocorticoids, and immunotherapy. It also spotlights intermittent high-dose steroids and ICI-related fractures, areas with scarce consolidated guidance. Clinically relevant for oncologists, geriatricians, and allied health professionals managing survivorship. Interest is therefore moderate-to-high despite limited novelty, because it addresses a common yet often under-prioritised issue with an aging cancer population.

1. Authors leveled the article type as “Commentary,” yet >8 000 words, 2 figures and 2 tables—reads as a narrative review. Re-classify as “Narrative Review” (or shorten drastically if true commentary).
2. Key points (e.g., fracture-risk tools, non-pharmacological advice) reiterated in ≥3 sections. Consolidate repeated content; move practical advice to one boxed algorithm.
3. Figure 1 copyright. Reproduces ESMO guideline flowchart almost verbatim without explicit permission note. 
4. Table 2 calculations. Conversions correct, but heading conflates daily-dose (NOGG) with intermittent cycles; mixing frameworks obscures risk interpretation. May lead clinicians to apply thresholds incorrectly. Separate daily-dose guidance from cycle-based examples; clarify cumulative‐dose rationale.
5. Statements such as “denosumab provides 50 % fracture reduction” given without GRADE level or study design qualifier. Add evidence levels (e.g., RCT, meta-analysis, observational) or at minimum “high-/moderate-quality evidence.”
6. No mention of anabolic agents (teriparatide, abaloparatide, romosozumab) now approved in many regions. 
7. Lacks discussion of osteosarcopenia—highly relevant to frail oncology patients.
8. Immunotherapy section. Acknowledges limited data but blends case-series with registry signals without hierarchy; conclusion (“overall risk may be low”) is speculative.
9. The manuscript’s citation practice does not comply with Cancers (MDPI) guidelines.
10. Occasional grammatical slips (“increase bone loss”, “treatment survivorship programmes”). Abstract is 29 lines—well above journal’s usual 10-15.
11. The manuscript notes FRAX is “not validated” but still recommends routine use without specifying necessary adjustments (e.g., secondary-osteoporosis tick box, steroid dose modifiers). Provide explicit guidance or at least caveat the limitations.
12. The choice of a 1 000 mg prednisolone cumulative threshold is extrapolated from immune-thrombocytopenia data, not oncology; acknowledges evidence gap but issues a quasi-guideline. Recast as expert opinion and highlight need for prospective studies.
13. The review cites a 14 % pooled RRIF rate yet later calls evidence “limited and inconclusive,” creating mixed messaging. Clarify that incidence is better characterised than intervention efficacy, and describe ongoing trials if any.
14. The flowchart (Figure 2) excludes patients with bone metastases, yet those individuals often receive identical agents (zoledronic acid, denosumab) for dual indications. Add a branch or explicit note to avoid confusion in mixed cohorts.

Author Response

Many thanks for the Reviewer for your comprehensive review and excellent suggestions for improvements in the manuscript. Attached are the Author's notes and revisions

The core theme—cancer-treatment-induced bone loss and its management—is not new; authoritative guidelines already exist (ASCO 2019, ESMO 2020). The manuscript’s value lies in bringing multiple guidelines under one roof and offering a practical care algorithm that spans chemotherapy, hormonal therapy, radiotherapy, glucocorticoids, and immunotherapy. It also spotlights intermittent high-dose steroids and ICI-related fractures, areas with scarce consolidated guidance. Clinically relevant for oncologists, geriatricians, and allied health professionals managing survivorship. Interest is therefore moderate-to-high despite limited novelty, because it addresses a common yet often under-prioritised issue with an aging cancer population.

  1. Authors leveled the article type as “Commentary,” yet >8 000 words, 2 figures and 2 tables—reads as a narrative review. Re-classify as “Narrative Review” (or shorten drastically if true commentary)

Response 1: The Autthors agree the length of the article is more suitable as a Narrative Review and will resubmitted as such.

  1. Key points (e.g., fracture-risk tools, non-pharmacological advice) reiterated in ≥3 sections. Consolidate repeated content; move practical advice to one boxed algorithm. Response 2: The Authors agree and have removed any repetition and consolidated these areas in the text
  2. Figure 1 copyright. Reproduces ESMO guideline flowchart almost verbatim without explicit permission note. 

Response 3 The Authors have consolidated the algorithms form Figure 1 and 2  into one single figure that is Author’s own and references the ESMO guidelines within this amended Figure.

  1. Table 2 calculations. Conversions correct, but heading conflates daily-dose (NOGG) with intermittent cycles; mixing frameworks obscures risk interpretation. May lead clinicians to apply thresholds incorrectly. Separate daily-dose guidance from cycle-based examples; clarify cumulative‐dose rationale.

Response 4. The Authors agree with your comments. We have separated the steroid dosing charts into two separate tables (Table 1 and Table 2) and clarified the commonly prescribes dosing regimes and potential implications

  1. Statements such as “denosumab provides 50 % fracture reduction” given

without GRADE level or study design qualifier. Add evidence levels (e.g., RCT, meta-analysis, observational) or at minimum “high-/moderate-quality evidence.”-

Response  5. The Authors agree with your comment. We have reviewed the evidence regarding the efficacy of denosumab and provided RCT and pooled analysis data in the text (Pg 4 172-183)

  1. No mention of anabolic agents (teriparatide, abaloparatide, romosozumab) now approved in many regions

Response 6. The Authors agree with your comment. We have added a section on anabolic agents, parathyroid hormone analogues are contraindicated in people with skeletal malignancies and Romosozumab has not been studied in cancer populations at the time of writing (Pg 12 442-450)

  1. Lacks discussion of osteosarcopenia—highly relevant to frail oncology patients.

Response 7. The Authors Agree and have added a discussion on osteosarcopenia and a recent meta-analysis (Pg 3 117-124)

  1. Immunotherapy section. Acknowledges limited data but blends case-series with registry signals without hierarchy; conclusion (“overall risk may be low”) is speculative.

Response 8. The Authors agree and have summarised the registry data and retrospective study in greater detail including GRADE evidence where applicable (Pg 9 340-360)

  1. The manuscript’s citation practice does not comply with Cancers (MDPI) guidelines.-

Response 9 The Authors agree and have updated the in text references to comply with Cancers Guidelines

  1. Occasional grammatical slips (“increase bone loss”, “treatment survivorship programmes”). Abstract is 29 lines—well above journal’s usual 10-15:

Response 10. The Authors agree the abstract was lengthy and have shortened the abstract to include the main points only ( Pg 1, 5-21) and have addressed grammatical errors

  1. The manuscript notes FRAX is “not validated” but still recommends routine use without specifying necessary adjustments (e.g., secondary-osteoporosis tick box, steroid dose modifiers). Provide explicit guidance or at least caveat the limitations.

Response 11. The Author’s agree- we have incorporated your suggestions and have highlighted the need and have caveated the limitations of the fracture score and necessary adjustments (Pg 9 370-377)

  1. The choice of a 1 000 mg prednisolone cumulative threshold is extrapolated from immune-thrombocytopenia data, not oncology; acknowledges evidence gap but issues a quasi-guideline. Recast as expert opinion and highlight need for prospective studies.

Response 12. The Author’s agree- we have incorporated your suggestions and have highlighted the need for robust data on this subject and re-framed the algorithm as Expert Opinion (Pg 8 312-321 and Pg 10 384-390)

  1. The review cites a 14 % pooled RRIF rate yet later calls evidence “limited and inconclusive,” creating mixed messaging. Clarify that incidence is better characterised than intervention efficacy, and describe ongoing trials if any.

Response 13. The Author’s agree- we have incorporated your suggestions on the wording around RRIF intervention and included an ongoing RCT (Pg 7 274-279)

  1. The flowchart (Figure 2) excludes patients with bone metastases, yet those individuals often receive identical agents (zoledronic acid, denosumab) for dual indications. Add a branch or explicit note to avoid confusion in mixed cohorts.

Response 14. The Authors agree- while bone targeted agents are often used in patients with bone metastasis, the dosing schedule is often more frequent and there are multiple other therapeutic options such as external beam radiation, vertebroplasty etc that should also may also considered which is outside the scope of this review. We have amended the Figure 1 to include the ESMO clinical guidelines for bone metastasis and included a reference (21) for bone targeted agents in metastatic disease for clarity

Reviewer 2 Report

Comments and Suggestions for Authors

An important issue the authors raise. I completely agree with the background shown in the Introdiction. 
However, I would like to add that the number of people with MRONJ/BRONJ is also growing intensively. In 2005-2015 the increase was low, but in the last decade this treatment has spread and the life expectancy of patients has increased, so the number of observed adverse effects has also increased. This is a minor complication as far as one concerned the main diagnosis in the patient, but it is very poorly treated in patients undergoing high-dose antiresorptive therapy (for example Denosumab or Bisphosphonate). And we are talking about elderly people in whom other deficits also occur in parallel.
I think the manuscript should state that before implementing antiresorptive therapy, the patient must be consulted by a dental surgeon and all foci of infection must be removed so that no oral surgery is needed for the next 5 years.

Author Response

An important issue the authors raise. I completely agree with the background shown in the Introdiction. 
However, I would like to add that the number of people with MRONJ/BRONJ is also growing intensively. In 2005-2015 the increase was low, but in the last decade this treatment has spread and the life expectancy of patients has increased, so the number of observed adverse effects has also increased. This is a minor complication as far as one concerned the main diagnosis in the patient, but it is very poorly treated in patients undergoing high-dose antiresorptive therapy (for example Denosumab or Bisphosphonate). And we are talking about elderly people in whom other deficits also occur in parallel.
I think the manuscript should state that before implementing antiresorptive therapy, the patient must be consulted by a dental surgeon and all foci of infection must be removed so that no oral surgery is needed for the next 5 years.

The Authors agree this was an important omission in the initial manuscript. We have included discussion on MRONJ including preventative dental management as suggested (Pg 11 422-435)

Many thanks to the Reviewer for your helpful comments and suggestions. 

Reviewer 3 Report

Comments and Suggestions for Authors

very interesting and important topic, due this reason, you have to write a systematyc review following PRISMA checklist to improve scientistic content and validity

Author Response

Comment: very interesting and important topic, due this reason, you have to write a systematyc review following PRISMA checklist to improve scientistic content and validity

Many thanks for your comment. The Author’s agree a systematic review would add significant value to the topic and will now be explored by the Authors in the near future as a follow up to this narrative review. Unfortunately due to length of time taken to register, perform and write a systematic review it is not possible to undertake a systematic review for this revision however we will undertake an SR as a separate project in due course.

Round 2

Reviewer 1 Report

Comments and Suggestions for Authors

All reviewer comments have been addressed satisfactorily. The authors have reclassified the article as a Narrative Review, consolidated repetitive content, redrawn figures to avoid copyright concerns, and corrected the steroid dosing tables. Evidence qualifiers for pharmacological agents such as denosumab were added, and new sections on anabolic agents, osteosarcopenia, and immunotherapy-related bone risks were incorporated. The FRAX guidance was appropriately caveated, the prednisolone threshold reframed as expert opinion, and the RRIF section clarified. References were reformatted to meet journal standards, the abstract was shortened, and grammatical issues were corrected. The revised flowchart now includes metastatic bone disease.

Author Response

Comment: All reviewer comments have been addressed satisfactorily. The authors have reclassified the article as a Narrative Review, consolidated repetitive content, redrawn figures to avoid copyright concerns, and corrected the steroid dosing tables. Evidence qualifiers for pharmacological agents such as denosumab were added, and new sections on anabolic agents, osteosarcopenia, and immunotherapy-related bone risks were incorporated. The FRAX guidance was appropriately caveated, the prednisolone threshold reframed as expert opinion, and the RRIF section clarified. References were reformatted to meet journal standards, the abstract was shortened, and grammatical issues were corrected. The revised flowchart now includes metastatic bone disease.

Response: Many thanks for your helpful suggestions which we feel were most helpful in framing the discussion around bone health and highlighting important considerations which were not present in the original manuscript. The Author's appreciate the time taken to perform an in-depth critical analysis of our paper

Reviewer 3 Report

Comments and Suggestions for Authors

your comments are raight, please add a limitation section explaing de disadvantadges ogf narratoive versus systemmatic teviews and planning fufure systematic reviews

Author Response

Comment:  your comments are raight, please add a limitation section explaing de disadvantadges ogf narratoive versus systemmatic teviews and planning fufure systematic reviews

Response: The Authors' agree it would be helpful to frame the rationale and outcome of the Narrative Review and future research gaps. We have added this to the conclusion section  (pg 13 539-544)